# Asymptomatic Carotid Atherosclerosis Cardiovascular Risk Factors and Common Hypertriglyceridemia Genetic Variants in Patients with Systemic Erythematosus Lupus

**DOI:** 10.3390/jcm10102218

**Published:** 2021-05-20

**Authors:** Marta Fanlo-Maresma, Beatriz Candás-Estébanez, Virginia Esteve-Luque, Ariadna Padró-Miquel, Francesc Escrihuela-Vidal, Monica Carratini-Moraes, Emili Corbella, Xavier Corbella, Xavier Pintó

**Affiliations:** 1Cardiovascular Risk Unit-Internal Medicine Department, Hospital Universitari de Bellvitge, L’Hospitalet de Llobregat, 08907 Barcelona, Spain; vesteve@bellvitgehospital.cat (V.E.-L.); fescrihuela@bellvitgehospital.cat (F.E.-V.); emilic@bellvitgehospital.cat (E.C.); xcorbella@bellvitgehospital.cat (X.C.); xpinto@bellvitgehospital.cat (X.P.); 2Institut d’Investigació Biomèdica de Bellvitge (IDIBELL), L’Hospitalet de Llobregat, 08907 Barcelona, Spain; apadro@bellvitgehospital.cat; 3Clinical Laboratory, Hospital Universitari de Bellvitge, 08907 Barcelona, Spain; mcarrattinim@bellvitgehospital.cat; 4CIBEROBN Fisiopatología de la Obesidad y Nutrición, Instituto de Salud Carlos III, 28220 Madrid, Spain; 5Faculty of Medicine and Health Sciences, Universitat Internacional de Catalunya, 08017 Barcelona, Spain; 6Faculty of Medicine, Universitat de Barcelona, 08007 Barcelona, Spain

**Keywords:** triglycerides, *CGKR* gene, Non-HDL-C, atherosclerosis, systemic lupus erythematosus

## Abstract

SLE is associated with increased cardiovascular risk. The objective of this study was to determine the prevalence of asymptomatic carotid atherosclerosis to analyze its relationship with dyslipidemia and related genetic factors in a population of patients with SLE. Seventy-one SLE female patients were recruited. Carotid ultrasound, laboratory profiles, and genetic analysis of the *ZPR1, APOA5*, and *GCKR* genes were performed. SLE patients were divided into two groups according to the presence or absence of carotid plaques. Patients with carotid plaque had higher plasma TG (1.5 vs. 0.9 mmol/L, *p* = 0.001), Non-HDL-C (3.5 vs. 3.1 mmol/L, *p* = 0.025), and apoB concentrations (1.0 vs. 0.9 g/L, *p* = 0.010) and a higher prevalence of hypertension (80 vs. 37.5%, *p* = 0.003) than patients without carotid plaque. The *GCKR* C-allele was present in 83.3% and 16.7% (*p* = 0.047) of patients with and without carotid plaque, respectively. The *GCKR* CC genotype (OR = 0.026; 95% CI: 0.001 to 0.473, *p* = 0.014), an increase of 1 mmol/L in TG concentrations (*OR* = 12.550; 95% CI: 1.703 to 92.475, *p* = 0.013) and to be hypertensive (*OR* = 9.691; 95% CI: 1.703 to 84.874, *p* = 0.040) were independently associated with carotid atherosclerosis. In summary, plasma TG concentrations, *CGKR* CC homozygosity, and hypertension are independent predictors of carotid atherosclerosis in women with SLE.

## 1. Introduction

Systemic lupus erythematosus (SLE) is a systemic autoimmune disease characterized by inflammation and tissue damage produced by an exaggerated response of the immune system, the binding of autoantibodies to different cells, and the deposition of antigen-antibody complexes in tissues. SLE most commonly presents in women of reproductive age, has a highly variable clinical course, and is associated with a 3-fold increased risk of premature death, mainly due to infection, renal impairment, or cardiovascular disease [1]. Even during inactive disease, SLE patients may have characteristic dyslipidemia that is known as the “lupus pattern” consisting of hypertriglyceridemia and decreased levels of high-density lipoprotein cholesterol (HDL-C) [2]. In active SLE, this “lupus pattern” may be aggravated by the presence of anti-lipoprotein lipase (LPL) antibodies and a consequent decrease in lipolysis that results in the accumulation of triglyceride-rich lipoproteins [3]. Furthermore, high-density lipoprotein (HDL) efflux capacity is significantly impaired, and the expression of apolipoprotein E (apoE) is increased [4].

The increase in cardiovascular risk in patients with SLE is not justified by traditional risk factors alone, and chronic inflammation may play a role [5]. In addition, other more subtle alterations of the structure and function of HDL particles such as those that have been identified in patients with other inflammatory diseases [6], and an increase in the proportion of small dense low-density lipoprotein (sd-LDL) particles, which is known as the atherogenic lipoprotein phenotype, may be involved [7].

To date, the role that genetic disorders may play in the origin of dyslipidemia and cardiovascular disease in SLE is not well defined. Numerous investigations have concluded that the majority of hypertriglyceridemia is polygenic in nature; that is, they involve the participation of different genes [8]. Polygenic hypertriglyceridemia is produced by the aggregation of different allelic variants. These genetic variations can occur in more than 30 genes related to triglyceride metabolism, with different penetrance and population frequency [9]. The genome-wide association study carried out by the Global Lipids Genetics Consortium identified the genetic variants with the greatest effect on triglyceride concentrations [10]. We have selected those variables with the greatest effect on TG concentration and in the development of clinical HTG [10]. In addition, a study conducted by our working group presented in a national congress concluded that, of these genetic variants, the most frequently related to hypertriglyceridemia in our population were those of the ZPR1, APOA5, and GCKR genes. As already explained, lupus atherogenic dyslipidemia is one of the etiological factors of cardiovascular disease in patients with SLE. Mild-moderate hypertriglyceridemia is a frequent finding among these patients. Therefore, it is very interesting to analyze whether there is any polygenic component that can be identified in a prevalent way in these patients [11].

The objective of this study was to know the prevalence of carotid atherosclerosis and to analyze its relationship with dyslipidemia and the related genetic factors in a population of female patients with SLE.

## 2. Materials and Methods

### 2.1. Study Population

Seventy-one female patients diagnosed with SLE, based on the revised American College of Rheumatology (ACR, Atlanta, GA, USA) classification criteria for SLE, were recruited from the Systemic Autoimmune Diseases Unit of the Hospital Universitari de Bellvitge (L’Hospitalet de Llobregat, Barcelona, Spain). The research protocol was approved by the Ethics Committee of the hospital, and all patients provided informed consent to participate in the study. The exclusion criteria were: history of a previous atherosclerotic cardiovascular event and autoimmune disease other than SLE, non-cardiovascular lower limb amputation, and absence of ultrasound (US) image of the carotid bifurcation. The following clinical information was collected for each study participant included:Demographic data: age and ethnicity;Cardiovascular risk factors as defined by the Spanish Society of Arteriosclerosis (SEA) [12]: smoking status, diabetes mellitus, body mass index (BMI), and family history of early cardiovascular disease. Arterial hypertension was defined as a systolic blood pressure of 140 mm Hg or more, or diastolic blood pressure of 90 mm Hg or more, or being on antihypertensive treatment. Dyslipidemia was defined as a concentration of LDL-C equal to or higher than 100 mg/dL, or as a Non-HDL-C concentration equal to or higher than 130 mg/dL or being on lipid-lowering treatment. Information on premature menopause before the age of 40 years was also collected [13];Drugs: antithrombotic medications, statins, immunosuppressants;Lifestyle: Mediterranean Diet questionnaire [14] and physical activity;SLE: onset and time of disease evolution, analytical immunological activity, SLE disease activity index scales (Systemic Lupus Erythematosus Disease Activity Index [SLEDAI]) [15] and Systemic Lupus International Collaborating Clinics/ACR Damage Index (SDI) [16], secondary antiphospholipid syndrome, history of immunosuppressive drugs and analytical parameters of inflammation;Laboratory data: serum glucose, lipid profile, liver and kidney parameters, hematology, coagulation, and immunological tests;Carotid US and ankle-brachial index (ABI).

### 2.2. Laboratory Data 

All the biochemical analyses were performed in plasma using a COBAS 711 automated analyzer (Roche Diagnostics^®^, Basel, Switzerland), while homocysteine levels were determined by enzymatic analysis using the Immulite 2000 XPi analyzer (Siemens Healthcare GmbH. Erlangen, Germany). Hematology values were measured by impedance flow cytometry (Sysmex S.L XN 900), and coagulation parameters were determined using the analyzer ACTLOP. Immunological analyses were performed using standard enzymatic techniques. For measurement, sd-LDL particles were isolated, and samples containing elevated triglyceride concentrations were processed [17]. Surfactant magnesium heparin was prepared as a solution of 150 U/mL of heparin-Na^+^ (Sigma Aldrich H3393), with 90 mmol/L of MgCl_2_ and 300 µL of magnesium heparin being added to each 300 µL of plasma and incubated for 10 min at 37 °C. The mixture was then kept at 0 °C for 15 min and centrifuged at 21,913 × *g* for 15 min at 4 °C (6K15 SIGMA centrifuge), after which lipoproteins with a density < 1.044 g/mL were precipitated. The supernatant contained HDL and sd-LDL particles with densities between 1.044 and 1.063 g/mL. The supernatants of HDL-C and total cholesterol were measured using a homogeneous assay using a Cobas 8000 modular analyzer (Roche^®^ Diagnostics GmbH, Mannheim, Germany) [18].

### 2.3. Genetic Analysis of Polymorphisms Involved in Triglyceride Metabolism

Functional variants of the *APOE* gene were genotyped using validated TaqMan™ MGB probes.

Genetic analysis of the allelic variants of the *ZPR1*, *APOA5*, and *GCKR* genes which most frequently determine the presence of polygenic hypertriglyceridemia [19] in the population of our area, was performed. The three possible genotypes were dichotomised in homozygotes so as not to present a risk allele (0) or risk allele carriers (either hetero or homozygous) c*724C>G (*ZPR1*); (0 = CC vs. 1 = G), c.56C>G (*APOA5*); (0 = CC vs. 1 = G), c.1337T>C (*GCKR*); (0 = CC vs. 1 = T).

### 2.4. Carotid Ultrasound and Ankle-Brachial Index

Certified vascular technologists measured the carotid US and the ABI using standardized protocols. Carotid US was performed to assess carotid intima-media wall thickness (cIMT) in the far wall of the common carotid artery (CCA) and to detect focal plaques in the extracranial carotid tree with the commercially available scanner (ACUSON Antares™ Siemens Medical Solutions USA, Inc.) US system using a 6 MHz linear array transducer. Plaque criteria were defined according to the Mannheim consensus as a focal protrusion in the lumen with a cIMT > 1.5 mm, a protrusion at least 50% greater than the surrounding cIMT, or an arterial lumen > 0.5 mm [20]. The cIMT was defined as the distance between the leading edge of the lumen–intima echo and the leading edge of the media–adventitia echo [21]. Normal cIMT values in Spanish women corresponding to the seventy-fifth percentile are 0.60 mm for <46 years, 0.84 mm for <65 years, and 1.02 for >65 years. At inclusion, cIMT measurements above these values were considered pathologic. When a plaque was observed in the region of the CCA measurements, the cIMT was not measured [22].

The ABI was performed after 5 min of resting in a supine position. Systolic blood pressure was measured in the brachial arteries and at the ankle level and on both sides of the body using an automatic waveform analyzer (Vascular Handheld Doppler Bidop 7 Hadeko^®^, Kawasaki, Japan). The ABI was calculated as the ratio of the ankle to brachial pressure. Ratios from >0.90 to <1.4 are considered normal, while ratios ≤ 0.9 indicate the presence of peripheral artery disease [23] and ratios ≥ 1.4 usually indicate the presence of arterial calcification, which is also associated with an increased risk of cardiovascular complications and increased risk of all-cause mortality [12,24].

### 2.5. Statistical Analyses

Qualitative variables were described as absolute frequency (percentage) and were analyzed using Chi-square or Fisher’s exact tests. Normally distributed quantitative variables were described as mean (standard deviation) and analyzed by analysis of variance (ANOVA). Variables that were not distributed normally were described as median (inter-quartile range) and analyzed with nonparametric tests (Mann–Whitney U). To analyze the relationship between genetic variables and the presence of plaque, a binary logistic regression model was carried out, with the presence of plaque as a dependent variable, genetic variables as independent variables, and adjusted with different covariates. The selected covariates were the significative variables in the bivariant analysis (hypertension and triglycerides, dyslipidemia, Non-HDL-C, and statins/ezetimibe treatments were not included due to the high correlation with the triglycerides) or almost significant variables (*p* < 0.1; age, SLE disease length). The clinical variables related to SLE severity: Systemic Lupus International Collaborating Clinics (SLICC) score, nephritis classification, and cumulative prednisone dose, were also included. A *p*-value cutoff *p* < 0.05 was considered significant.

## 3. Results

### 3.1. Subjects Characteristics

Seventy-one female SLE patients met the inclusion criteria and were enrolled in the study. As shown in Table 1, the mean age of the subjects at enrollment was 52.2 years (30–75) with a median BMI of 25.5 kg/m^2^ and a mean waist circumference of 88.4 cm (58–129). Thirty women (42.3%) had a family history of cardiovascular disease, 46.5% were hypertensive, 53.5% had dyslipidemia, and 7% were diabetic. The mean time of SLE disease duration was 20.7 years. Antiphospholipid syndrome was present in 14.1% of subjects. Thirty-four (47.9%) had a history of lupus nephritis. Regarding treatments, 46.4% of patients were on statins and 28.1% on antithrombotic drugs. Sixty-five (91.5%) women were on hydroxychloroquine. Sixty-three (88.7%) subjects had been treated with corticosteroids at some point after SLE diagnosis.

### 3.2. Comparisons between Subgroups of SLE-Patients

Two groups of SLE patients were considered according to the presence or absence of carotid plaques. Carotid plaques were present in 15 patients (21.1%). No significant differences were observed between the two groups in regard to age, race, educational level, dietary pattern, smoking, and alcohol consumption. There were also no significant differences in relation to early menopause and other gynaecologic-obstetric aspects, including contraceptive use, menarche, early menopause, pregnancies, miscarriages, and breastfeeding, or with respect to factors directly related to SLE, such as antiphospholipid syndrome, nephritis, and SLE disease activity assessed with the SLEDAI and SLICC scores.

Compared to patients without carotid plaque, SLE patients with carotid plaque had higher concentrations of plasma triglycerides (1.5 vs. 0.9 mmol/L, *p* = 0.001), non-HDL-C (3.5 vs. 3.1 mmol/L, *p* = 0.025), apolipoprotein B (apoB) (1.0 vs. 0.9 g/L, *p* = 0.010) and homocysteine (13 vs. 10.5, *p* = 0.037). Patients with carotid plaque also had a higher prevalence of hypertension than those without carotid plaque (80% vs. 37.5%, *p* = 0.003), as well as a higher prevalence of dyslipidemia (86.7% vs. 44.6%, *p* = 0.004). The clinical course of SLE was longer, albeit not statistically significant, in patients with carotid plaque than in those without (24.1 vs. 19.8 years, *p* = 0.063). As shown in Table 1, cholesterol-lowering drugs were more frequently used by SLE patients with carotid plaque than by SLE patients without carotid plaque: statins 86.7% vs. 35.7%, respectively (*p* < 0.001) and ezetimibe 20% vs. 0%, respectively (*p* = 0.008).

Treatment with immunosuppressant drugs was also more frequent in SLE patients with carotid plaque than in SLE patients without carotid plaque: methotrexate 33.3% vs. 7.3%, respectively (*p* < 0.018) and other immunosuppressants including cyclophosphamide, azathioprine, and rituximab, 60% vs. 38.2%, respectively (*p* = 0.130). Among patients with carotid plaque, 100% had received corticosteroids at some point in the disease compared to 85.7% of patients without carotid plaque (*p* < 0.189). Eight patients had never received glucocorticoid treatment. The patients were divided according to tertiles of cumulative prednisone doses of 0–17 g, >17–51 g, and >51 g, and the presence of carotid plaque was evaluated by groups. No significant differences were found between the different cumulative prednisone groups (Table 2).

Nor were significant differences found when studying the different subgroups of lupus nephritis (Table 3).

The results of the genetic analyses of the three allelic variants that most commonly influence plasma triglycerides concentrations of patients in our area are shown in Table 4.

Comparable proportions of subjects had carotid plaque across the ZPR1 and APOA5 genotypes. Patients who were homozygote and heterozygote for a protective GKCR C-allele had a lower prevalence of carotid plaque than homozygote patients with the GCKR (c.1337C>T) TT variant (16.7% vs. 45.5%, *p*= 0.047).

In the multivariate logistic regression analysis (Table 5) it was observed that the protective GCKR rs1260326 CC genotype reduced the risk of having carotid plaque (OR = 0.026; 95% CI: 0.001 to 0.473, *p* = 0.014). Furthermore, a one millimole per liter increase intriglycerides was associated with an increased risk for carotid plaque (OR = 12.550; 95% CI: 1.703 to 92.475, *p* = 0.013) as well as hypertension (OR = 9.691; 95% CI: 1.703 to 84.874, *p* = 0.040). The whole model explained 56.2% (Nagelkerke R2) of the risk for developing carotid plaque in our SLE patients.

## 4. Discussion

To the best of our knowledge, this is the first study to link common genetic variants associated with plasma triglyceride concentrations to the presence of subclinical carotid atherosclerosis in SLE.

Detection of carotid plaques by carotid US is a strong predictor of future cardiovascular events in patients with SLE. More specifically, the presence of carotid plaque at baseline has been associated with a greater than 4-fold increased risk for any severe cardiovascular event in SLE patients [25]. Although intimal thickening has also been linked to increased cardiovascular risk, it is well established that the presence of atherosclerotic plaque is a better surrogate marker of cardiovascular risk than cIMT also in SLE patients [26,27,28,29]. In this sense, we only analyzed data on the presence of carotid plaque, which in our SLE population was found to be similar to what has been reported in previous series of SLE patients (21.1% vs. 16% to 37%) [30,31]. Patients with carotid plaque had a higher prevalence of hypertension and dyslipidemia than those without plaque, a finding that is consistent with the role of these are conventional cardiovascular risk factors in patients with SLE [32,33,34,35,36,37]. In line with previous results, patients with carotid plaque showed higher plasma triglyceride concentrations [38]. Although non-HDL-C and apoB values were higher in patients with plaque, there were no statistically significant differences in low-density lipoprotein cholesterol (LDL-C). Non-HDL-C includes the atherogenic potential of LDL-C together with the cholesterol of remnant lipoproteins and provides a more accurate estimation of risk than LDL-C, especially in subjects with hypertriglyceridemia [39]. These results are in agreement with current knowledge of the role of triglyceride-rich lipoproteins and their remnants in the origin and progression of atherosclerosis, which is strongly influenced by the number of circulating concentrations of apoB-containing lipoprotein particles [40]. On the other hand, HDL-C concentrations were not significatively different in patients with and without carotid atherosclerosis. This result is in agreement with the data from Mendelian randomization studies that do not provide compelling evidence that HDL-C is causally associated with the risk of atherosclerotic cardiovascular disease [41].

The study of sd-LDL in SLE patients with atherosclerosis has been addressed in different investigations. With the use of the Lipoprint LDL system, it has been demonstrated that LDL particles are smaller and denser in patients with SLE than in healthy controls [42]. Another study reported that circulating lipoprotein remnant particles measured by magnetic resonance were better predictors of carotid atherosclerosis in SLE subjects than other lipoprotein variables [42]. In spite of the higher triglyceride and non-HDL-C plasma concentrations found in SLE patients with carotid atherosclerosis in the present study, there were no differences in sd-LDL concentrations that were measured with the modified heparin-Mg^2+^ precipitation method [22] between SLE patients with and without carotid atherosclerosis. The association of the ε_2_ APOE allele with increased cIMT in SLE patients has also been suggested [42], but this was not observed in SLE patients from this study.

Triglycerides are carried in plasma by very-low-density lipoproteins (VLDL), chylomicrons, and their remnants, which are highly atherogenic. There is a direct relationship between plasma triglyceride and cholesterol concentrations in remnant particles. There is strong evidence suggesting that elevated triglyceride-rich lipoproteins are causal risk factors for low-grade inflammation and atherosclerosis [43,44]. Mild-to-moderate hypertriglyceridemia is a polygenic disorder, which is the consequence of a cumulative burden of common and rare genetic variants that are usually exacerbated by non-genetic factors [9]. In this study, the analysis of the *ZPR1, APOA5*, and *GCKR* genes in patients with SLE, showed that *GCKR* rs1260326 is strongly related to the presence of carotid atherosclerosis. The glucokinase regulator *GCKR* gene encodes the glucokinase regulatory protein (GKRP), which is released in the postprandial phase to the cytoplasm and stimulates glycogen deposition and *de novo* lipogenesis. The presence of the homozygous TT allele of *GCKR* rs1260326 produces destabilization of the glucokinase binding interface. Increased hepatic glucokinase activity results in higher fasting serum triglyceride and lower glucose concentrations [45,46]. In a meta-analysis of 46 genome-wide association studies a significant association was observed between the homozygous TT allele of *GCKR* rs1260326 and higher circulating triglyceride levels [47], although the atherogenic effect of this variant is controversial [48,49,50]. Evaluation of non-traditional cardiovascular risk factors revealed statistically significant differences in plasma homocysteine concentrations in patients with carotid plaque (*p* = 0.037), which is consistent with the reported role of homocysteine as a potential contributor to the increased burden of atherosclerotic disease in SLE [51,52]. Antiphospholipid antibodies and SLE nephritis are tightly related to the risk of cardiovascular events in SLE patients [30,32,53,54], although neither antiphospholipid syndrome nor lupus nephritis or the complement, were statistically relevant in our population.

Regarding the severity and chronicity of SLE, the average SLEDAI score of disease activity was moderate (4 of 2 to 6). Nevertheless, the women included in this study had less accumulated structural damage evaluated using the SLICC index than those described in former cardiovascular risk reports, and the length of SLE disease was longer (20.7 years) than in other investigations. The length of SLE disease evolution may be correlated with a long time of chronic inflammation, although it has not been proposed as a strong risk factor for the appearance of cardiovascular events [55,56].

In the present study, multivariate logistic regression was performed to identify the independent predictors of carotid atherosclerosis in patients with SLE and showed that *CGKR* CC homozygosity for the *CGKR* gene, plasma triglycerides, and hypertension (*OR* = 0.026, 12.550 and 9.691, respectively) remained independently associated with the development of plaque. Remarkably, the risk of having carotid plaques was directly related to the highest plasma triglyceride concentrations, despite the mean triglyceride concentrations being within the reference intervals (<150 mg/dL or <1.7 mmol/L). As reported, there was a direct relationship between triglyceride levels and mortality in patients with established coronary heart disease, even in patients with plasma triglyceride levels < 150 mg/dL [57].

Therefore, it should be considered whether in patients with SLE or other autoimmune or chronic inflammatory diseases that are associated with an alteration in triglyceride metabolism, a more comprehensive treatment of dyslipidemia could be recommended. Thus, in addition to statins to lower the cholesterol carried by atherogenic lipoproteins, they could also be treated with drugs aimed at the treatment of hypertriglyceridemia, which at this time are limited to fibrates and omega-3 fatty acids. In this sense, the European guidelines for the management of dyslipidaemias indicate that in patients at high cardiovascular risk who are treated with statins, the association with n-3 PUFAs (icosapent ethyl 2 × 2 g/day) should be considered when triglyceride concentrations are >135 mg/dL, but the association of statins with fibrates (fenofibrate or bezafibrate) may be considered only with triglyceride levels > 200 mg/dL [41].

The association between hypertension and atherosclerosis in the general population and in SLE patients is well-known [54]. Although a history of nephritis and SLE disease duration and activity (SLICC) were included in the multivariate statistical model because of their clinical relevance, their predictive power of carotid atherosclerosis was lost when the *CGKR* genetic study, triglycerides, and hypertension were included in the model. Corticosteroid treatment is one of the most important factors related to cardiovascular events in SLE patients in a dose and time-dependent manner [36,37,58], and other immunosuppressive agents such as cyclophosphamide or azathioprine have also been related to cardiovascular risk in these patients [32,35]. However, no significant differences were found when glucocorticoids were included in the multivariate model. In fact, the use of low-dose glucocorticoids has a favorable effect on the lipid profile [59].

This study had some limitations. Only women were included, and since SLE usually affects adult women in a ratio of 6:1 to men, it was decided not to include men in the study due to the small sample size [60]. In addition, a model adjusted for SLE and dyslipidemia treatments would have given more precise information.

The higher prevalence of statin treatment in SLE subjects with carotid plaque may be the consequence of the worse lipid profile of patients with atherosclerosis and also the more intensive lipid-lowering therapy that is indicated in these patients. Despite atherogenic dyslipidemia being a well-demonstrated cardiovascular risk factor [61], the mechanisms by which the treatment of hypertriglyceridemia and associated lipoprotein disorders reduces cardiovascular disease have not yet been fully established. Currently, there are several ongoing clinical trials to assess the benefit of hypertriglyceridemia treatment in cardiovascular disease [62]. It would be interesting to verify the protective cardiovascular effect of these drugs in lupus patients with asymptomatic atherosclerosis and atherogenic dyslipidemia in which triglyceride-rich lipoproteins play a major role. Finally, other limitations of this study that should be taken into account are the cross-sectional design and its sample size. Probably, some effects of the other risk factors that have been mentioned previously might have been missed due to the limited size of the cohort. Studies with larger sample sizes are needed to deep in our understanding of the relationship between immunosuppressive treatments or LN with atherosclerosis in patients with SLE.

## 5. Conclusions

This study shows that *CGKR* CC homozygosity for the *CGKR* gene, plasma triglyceride concentrations, and hypertension are independent predictive factors of carotid atherosclerosis in women with SLE. These data suggest that adequate control of hypertriglyceridemia and hypertension may be useful to prevent atherosclerosis in SLE. Finally, the results of this study also suggest that the diagnosis of genetic variants related to hypertriglyceridemia may be useful to better stratify cardiovascular risk in patients with SLE.

## Figures and Tables

**Table 1 jcm-10-02218-t001:** Baseline characteristics of the patients included in the study.

Variables	Total (*n* = 71)	No Carotid Plaque (*n* = 56)	Carotid Plaque (*n* = 15)	*p-*Value
Age (years)	52 (9.9)	51 (10.3)	56 (7.3)	0.067
BMI (kg/m^2^)	25.5 (23.0 to 29.6)	25.4 (22.9 to 29.0)	25.9 (23.2 to 30.7)	0.460
Waist circumference (cm)	88.4 (12.8)	88.0 (12.2)	90.0 (15.2)	0.611
Dietary questionnaire SEA (score)	11 (9 to 12)	11 (9 to 12)	11 (10 to 12)	0.585
Smoking (pack/years)	0,9 (0 to 15.7)	3.0 (0 to 15.9)	0 (0 to 14.5)	0.268
Hypertension	33 (46.5%)	21 (37.5%)	12 (80%)	**0.003**
Diabetes mellitus	5 (7%)	5 (8.9%)	0	0.577
Dyslipidemia	38 (53.5%)	25 (44.6%)	13 (86.7%)	**0.004**
CV familial history	30 (42.3%)	21 (37.5%)	9 (60%)	0.117
APS	10 (14.1%)	8 (10.7%)	4 (26.7%)	0.202
Nephritis	34 (47.9%)	24 (42.9%)	10 (66.7%)	0.101
SLE disease length (years)	20.7 (8.1)	19.8 (7.2)	24.1 (10.1)	0.063
Pathological ABI	5 (7%)	5 (8.9%)	0	0.577
Corticosteroids	63 (88.7%)	48 (85.7%)	15 (100%)	0.189
Metrotexate	9 (12.9%)	4 (7.3%)	5 (33.3%)	**0.018**
Other immunosuppressants	30 (42.9%)	21 (38.2%)	9 (60%)	0.130
GFR CKD-EPI (mL/min/1.73 m^2^)	90 (78 to 90)	90 (84.5 to 90)	89 (64 to 90)	0.068
TC (mmol/L)	4.9 (0.7)	4.8 (0.7)	5.2 (0.8)	0.091
LDL-C (mmol/L)	2.6 (0.6)	2.6 (0.6)	2.8 (0.6)	0.431
HDL-C (mmol/L)	1.7 (0.4)	1.7 (0.5)	1.6 (0.4)	0.489
TG (mmol/L)	1.0 (0.8 to 1.4)	0.9 (0.7 to 1.3)	1.5 (1.1 to 2.1)	**0.001**
Non-HDL-C (mmol/L)	3.2 (0.7)	3.1 (0.7)	3.5 (0.7)	**0.025**
ApoB (g/L)	0.9 (0.8 to 1.0)	0.9 (0.8 to 1.0)	1.0 (0.9 to 1.1)	**0.010**
Sd-LDL (mmol/L)	1.2 (0.3)	1.2 (0.3)	1.2 (0.4)	0.572
Lipoprotein (a) (g/L)	0.17 (0.07 to 0.47)	0.17 (0.07 to 0.37)	0.25 (0.05 to 1.30)	0.430
Statins	33 (46.5%)	20 (35.7%)	13 (86.7%)	**<0.001**
Ezetimibe	3 (4.2%)	0	3 (20%)	**0.008**
Homocysteine (μmol/L)	11 (9 to 14.9)	10.5 (8 to 14.5)	13 (11 to 18)	**0.037**
SLICC (score)	0 (0 to 1)	0 (0 to 1)	0 (0 to 1)	0.578
SLEDAI (score)	4 (2 to 8)	5 (2 to 8)	4 (2 to 6)	0.083
C3 complement (mg/L)	1049 (233)	1050 (241)	1042 (212)	0.904
C4 complement (mg/L)	172 (84)	175 (88)	162 (71)	0.601
Prothrombin time (ratio)	1.0 (0.9 to 1.0)	1.0 (0.9 to 1.0)	1.0 (0.9 to 2.1)	0.703
Fibrinogen (g/L)	3.2 (2.7 to 3.7)	3.2 (2.6 to 3.7)	3.4 (2.8 to 3.8)	0.489

Data are expressed as *n* (%); mean (standard deviation) for normally distributed quantitative variables, median (interquartile interval) for non-normally distributed variables, and chi^2^ test or Fisher test for qualitative variables. Value data highlighted in bold indicate *p* < 0.05. BMI: body mass index; SEA: Spanish Society of Atherosclerosis; CV: cardiovascular; APS: antiphospholipid syndrome; SLE: systemic lupus erythematosus; ABI: ankle brachial index; GFR: glomerular filtration rate; TC: total cholesterol; LDL-C: low-density lipoprotein cholesterol; HDL-C: high-density lipoprotein cholesterol; TG: triglycerides; Non-HDL-C: non-HDL-cholesterol; Apo B: apolipoprotein B; sd-LDL: small dense LDL; SLICC: Systemic Lupus International Collaborating Clinics; SLEDAI: Systemic Lupus Erythematosus Disease Activity Index. Other immunosuppressants were cyclophosphamide, azathioprine, rituximab, and mycophenolate.

**Table 2 jcm-10-02218-t002:** Cumulative prednisone classification.

Glucocorticoids (CPD)	No Carotid Plaque (*n* = 56)	Carotid Plaque (*n* = 15)	Total	*p-*Value
0 to ≥17 g	19 (33.9%)	4 (26.7%)	23 (32.4%)	0.182
>17 g to ≤51 g	21 (37.5%)	3 (20.0%)	24 (33.8%)	
>51 g	16 (28.6%)	8 (53.3%)	24 (33.8%)	

Data are expressed as *n* (%). CPD = cumulative prednisone dose.

**Table 3 jcm-10-02218-t003:** Lupus nephritis.

LN	No Carotid Plaque (*n* = 56)	Carotid Plaque (*n* = 15)	Total	*p-*Value
No history of LN	32 (57.1%)	5 (33.3%)	37 (52.1%)	0.162
Class I/II	4 (7.1%)	3 (20.0%)	7 (9.9%)	
Class III-V	20 (35.7%)	7 (46.7%)	27 (38.0%)	

Data are expressed as *n* (%). LN: lupus nephritis.

**Table 4 jcm-10-02218-t004:** Comparison of carotid plaque across genotypes.

Gene	Allelic Variant	No Carotid Plaque	Carotid Plaque	Total	*p-*Value
*ZPR1* (c.*724C>G)	CC	38 (76%)	12 (24%)	50	0.617NA
	CG	17 (85%)	3 (15%)	20	
	GG	1 (100%)	0 (0%)	1	
Grouped *ZPR1* (c.*724C>G)	CC or CT	55 (78.6%)	15 (21.4%)	70	>0.999
	TT	1 (100%)	0 (0%)	1	
*APOA5* (c.56C>G)	CC	46 (78%)	13 (22%)	59	0.838NA
	CG	9 (81.8%)	2 (18.2%)	11	
	GG	1 (100%)	0 (0%)	1	
Grouped *APOA5* (c.56C>G)	CC or CT	55 (78.6%)	15 (21.4%)	70	>0.999
	TT	1 (100%)	0 (0%)	1	
*GCKR* (c.1337C>T)	CC	23 (85.2%)	4 (14.8%)	27	0.069NA
	CT	27 (81.8%)	6 (18.2%)	33	
	TT	6 (54.5%)	5 (45.5%)	11	
Grouped *GCKR* (c.1337C>T)	CC or CT	50 (83.3%)	10 (16.7%)	60	**0.047**
	TT	6 (54.5%)	5 (45.5%)	11	

Data are expressed as *n* (%), and value data highlighted in bold indicate *p-*values *p* < 0.05. NA: not applicable (not applicable due to breach of statistical application conditions).

**Table 5 jcm-10-02218-t005:** Multivariate analysis of the risk factors for carotid plaque in SLE patients.

Variables	OR (IC 95%)	*p*-value
*GCKR* (c.1337C>T)-TT	Ref.	
-CC	0.026 (0.001 to 0.473)	0.014
-CT	0.119 (0.010 to 1.389)	0.090
Triglycerides (mmol/L)	12.550 (1.703 to 92.475)	0.013
Hypertension	9.691 (1.703 to 84.874)	0.040
Lupus nephritis-No	Ref.	
-class I/II	2.979 (0.186 to 47.785)	0.441
-class III-V	2.780 (0.439 to 17.590)	0.277
CPD (g)	0.977 (0.942 to 1.013)	0.213
SLE disease length (years)	1.099 (0.974 to 1.240)	0.124
SLICC (score)	1.045 (0.613 to 1.782)	0.871
Age (10 years)	2.125 (0.878 to 5.142)	0.094

OR: odds ratio; CI: confidence interval. CPD: cumulative prednisone dose; SLE: systemic lupus erythematosus; SLICC: Systemic Lupus International Collaborating Clinics. Age (10 years) indicates ten years age difference. Value data highlighted in bold indicate *p-*values *p*< 0.05.

## Data Availability

The data presented in this study are available on request from the corresponding author. The data are not publicly available due to respect the privacy of individuals.

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
