# Peer review of "Asymptomatic Carotid Atherosclerosis Cardiovascular Risk Factors and Common Hypertriglyceridemia Genetic Variants in Patients with Systemic Erythematosus Lupus"

_jcm, 2021, doi:10.3390/jcm10102218_

Round 1
Reviewer 1 Report
Although the topic is interesting and cross-sectional there are several methodological issues, which limit the significance of the findings:
- Over 46% of patients were on statins (however data is not presented in the Table 1 as stated in the text!) which strongly affects the lipid profile (one of the analyzed variables).
- There are no data on treatment with immunosuppressive drugs, which can affect the presence of carotid plaques.
- It is not explained why some variables and others not were chosen for multivariable analysis. What was the criterium? Was it arbitrary? P-value in univariate analysis?
- The studied genes and previous data on this subject should be briefly highlighted in the introduction section.
- How were hypertension and dyslipidemia defined? It is not clear
- There is no control group of women without SLE of similar age.
- Line 71 - such (as) those
- Line 197 - with plaque is missing
Author Response
Dear reviewer,
Thank you very much for your comments.
Please, see the attachment with the response.
Best regards,
Marta Fanlo Maresma

Reviewer 2 Report
This study analysed the prevalence of asymptomatic carotid atherosclerosis and its association with dyslipidemia and related genetic factors in a cohort of 71 female patients with SLE. Plasma triglyceride concentrations, CGKR CC homozygosity and hypertension were found to be independent predictors of carotid atherosclerosis.
This is a clear, well presented study. I have only minor concerns:
- The discussion would benefit from adding some idea about what type of therapy (to lower TG levels) the authors would suggest. More systematic use of statins (considering ‘negative’ data on the use of atorvastatin in SLE) ? others ?
- Some effects of other risk factors (that were discussed) might have been missed due to the limited size of the cohort (in particular when compared to classical cardiovascular studies). This limitation is shortly mentioned but the statement should be more clear, in particular as there was (as expected) numerically more LN in the patients with carotid artery plaque and more frequently a high steroid burden. These differences could be meaningful even if not significant. Maybe repetition in larger cohort study effort required in the future.
- Why did the authors exclude male patients ? (for which the situation could be somewhat different). This issue should also be part of the discussion.
Author Response

(The authors gave the same response as above.)

Round 2
Reviewer 1 Report
Thank you for the improvements in the manuscript. Most of my comments were addressed. I have few more remarks:
- If treatment with methotrexate and statins/ezetimibe was shown to have significantly different frequency between those with and without carotid plaques why was it not included in the multivariate analysis (and cumulative prednisone dose which was not significant in binary analysis was)?
- "Dyslipidemia was defined as a concentration of LDL-C equal to or higher than 100 mg/dL, or as a Non-HDL-C concentration equal to or higher than 130 mg/dL or more." - or being on hypolipemic treatment should be added.
- In the statistical section it should be almost significant variables p<0.1 instead of p<0.001)
Author Response
Dear Editor,
Thank you for this second review of our manuscript entitled "Asymptomatic Carotid Atherosclerosis Cardiovascular Risk Factors and Common Hypertriglyceridemia Genetic Variants in Patients with Systemic Erythematosus Lupus” and for giving us the opportunity to submit a second revised and improved version.
Below you will find a point-by-point response to the comments and questions of the reviewers.
In addition, we want to inform you that we have added the following sentence in authors contribution: “Xavier Corbella and Xavier Pinto contributed equally as senior authors” (line 410)
Sincerely,
Marta Fanlo Maresma
